# Investigation into Thermomechanical Response of Polymer Composite Materials Produced through Additive Manufacturing Technologies

**DOI:** 10.3390/ma15145069

**Published:** 2022-07-21

**Authors:** Raluca Maier, Anca Mihaela Istrate, Alexandra Despa, Andrei Cristian Mandoc, Sebastian Bucaciuc, Romică Stoica

**Affiliations:** Romanian Research & Development Institute for Gas Turbines—COMOTI, 061126 Bucharest, Romania; anca.istrate@comoti.ro (A.M.I.); alexandra.despa@comoti.ro (A.D.); andrei.mandoc@comoti.ro (A.C.M.); sebastian.bucaciuc@comoti.ro (S.B.); romica.stoica@comoti.ro (R.S.)

**Keywords:** additive manufacturing, FDM/FFF, CFF, DLP, tensile, three-point bending TMA, DMA, thermomechanical analysis

## Abstract

This paper presents the static mechanical behavior and the dynamic thermomechanical properties of four market-available reinforced and non-reinforced thermoplastics and photopolymer materials used as precursors in different additive manufacturing technologies. This article proposes a characterization approach to further address development of aeronautic secondary structures via 3D-printed composite materials replacing conventional manufactured carbon fiber reinforced polymer (CFRP) composites. Different 3D printing materials, technologies, printing directions, and parameters were investigated. Experimental results showed that carbon-reinforced ONYX_R material exhibits a transition point at 114 °C, a 600 MPa tensile strength, and an average tensile strain of 2.5%, comparable with conventional CFRP composites manufactured via autoclave, making it a suitable candidate for replacing CFRP composites, in the aim of taking advantage of 3D printing technologies. ONYX material exhibits higher stiffness than Acrylonitrile-Butadiene-Styrene Copolymer (ABS), or conventional Nylon 6/6 polyamide, the flexural modulus being 2.5 GPa; nevertheless, the 27 °C determined transition temperature limits its stability at higher temperature. Daylight High Tensile (further called HTS) resin exhibits a tensile strength and strain increase when shifting the printing direction from transversal to longitudinal, while no effect was observed in HighTemp DL400 resin (further called HTP).

## 1. Introduction

Industrial production is currently driven by global competition and the need for fast adaptation of production. Innovation and technological development are at the core of the economic growth process, with industry evolution being linked to cutting-edge research outputs related to manufacturing processes, materials, and product design. These requirements can be met by radical advances in traditional manufacturing technology or, as it was seen over the last 10 years, by the raising of emerging additive manufacturing technologies that present continuous growth and represent high interest for engineering due to undeniable major advantages related to increased automation level, production of complex geometries, low material usage, and design freedom. Understanding the key principles of each mainstream AM process is essential to design parts, products, and business strategies that leverage AM. More concretely, this information enables the user to select AM processes for specific applications, and design-integrated operations (e.g., including printing and post-processing), to meet application specific needs. AM methods have been used in different industries such as the aerospace industry [1,2], medical applications [3], automobile industries [1,2], construction [4], and so forth. The increased use of 3D printing as a learning tool and to generate functional end-use parts has brought out the need for a better understanding of the thermomechanical behavior of 3D-printed parts and the development of analytical tools and design guidelines for engineers [5,6]. There is a broad spectrum of AM processes, materials, and related technologies. The ISO/ASTM 52900 [7] standard ranks additive manufacturing technologies into seven classes. Additive manufacturing does not refer strictly to the 3D printing process. The processes that are part of additive manufacturing involve the post-processing and heat treatment of the components. Moreover, for reinforced composite materials intensively used in the aerospace industry, there is some complexity in the certification of these components developed using AM technologies due to their undeniable advantages in simplifying the manufacturing stages, compared to conventional methods (e.g., autoclave or VA-RTM). In this frame, two AM technologies that are gaining ground in the aerospace industry are FFF/CFF technology that brings continuous fiber reinforcement to the plastic matrix 3D-printed parts, exhibiting benefits of shrinkage [8] and mechanical strength close to the aluminum used in aerospace [9], and the DLP technology which offers rapid prototyping with great mechanical characteristics.

This study focuses on the two abovementioned AM technologies. The Markforged X7 has, in addition to the capacity to print thermoplastics, a second nozzle that adapts the CFF process to print non-plastics. In CFF manufacturing, an FFF printer with a second nozzle places continuous carbon fiber, fiberglass, or Kevlar^®^ in one final part, but instead of melting the entire filament, the heat of its nozzle is used for “integration” into the thermoplastic layer. The fibers do not melt; instead, they are captured by the thermoplastic matrix in a similar way to thermoset-type adhesives, as the fibers are captured by the epoxy matrix in traditional methods of making composite laminates. Turner et al. [10,11] provide an extensive review on FFF process modelling, including the flow and thermal dynamics of the melt, the extrusion process, and the bonding process between successive layers of material. Temperature, viscosity, and surface energy of the melt play an important role in how the material flows through the nozzle and, more importantly, how the final interface between the beads is formed. One of the major process variables is the raster angle, which leads to different properties across the principal material directions [12,13,14], similar to the orthotropic behavior of fiber composites.

Latest studies on the technology offered by Markforged focus on the mechanical characterization of the thermoplastic (ONYX) and mainly the continuous carbon fiber and glass fiber reinforcement and the effect that infill geometry has on the final 3D-printed part [5,6,7,8,15,16].

Less of the encounter research focused on specific applications for aerospace. Chen et al. [17] focused on using Markforged technology to transition from traditional manufacturing of molding processes to 3D printing such molds for an antenna. Their finding suggests that the coefficient of thermal expansion (CTE) of their 3D-printed configuration is mostly anisotropic due to the bad adhesion between 3D-printed layers and the aligned short carbon fiber along the printing pattern. They further suggested an inclined mode of 3D printing that could uniform the CTE of the antenna mold and solve the problems of larger printing steps. Jacob Chekal [9] focused his work on identifying and designing a new service access door for a Boeing airliner based on 3D printing materials available. As for the DLP 3D printing, the investigated research papers follow mostly a transition of AM to obtain 3D-printed injection molds. Lozano et al. [18] offered a thorough review, whose main findings concern polymer molds obtained through AM, in comparison to conventional (metal) molds obtained by subtractive manufacturing. They reported that information on specific topics is scarce or nonexistent. Such an example is on the characterization of the most commonly injected materials and molds used in this type of technology, their mechanical properties, for both the part and the mold and even a lack of the designs for all types of geometries, and costs.

This paper’s aim is to take the two technologies and further investigate their potential in the additive manufacturing of aeronautic secondary structures, such as OGVs (outer guiding vanes) and substitute conventional molds for use in the manufacturing of CFRP via autoclave technology; both applications require high thermal dimensional stability and mechanical strength. As the photopolymers used in 3D printing are different from one manufacturer to another, there is an obvious opportunity to investigate this matter. The Photocentric LC Magna 3D printer stays at the core of the DLP technology investigation of this paper, the manufacturer providing a photopolymer resin that can maintain a good thermomechanical behavior up to 230 °C. At the same time, the Markforged technology has the potential of providing the means to develop aeronautic secondary structures such as OGVs and molds up to a temperature of 140 °C. The study focuses on obtaining the required thermomechanical behavior for such 3D-printed parts by investigating a sandwich-like internal structure (ONYX/carbon fiber/ONYX) which can be further optimized by increasing the number of ONYX and carbon fiber interlayers.

## 2. Materials and Methods

### 2.1. Materials and Fabrication Methods

The materials used from Photocentric in the case of LCD printing for experimental test samples were HighTemp DL400 resin (further called HTP) and Daylight High Tensile (further called HTS) resin. In the case of FDM technology, the material used was ONYX (further called ONYX) while for CFF printing technology, carbon-fiber-reinforced ONYX (further called ONYX_R) was used, both supplied by Markforged (Watertown, MA, USA). For a comparative analysis, flexural and tensile specimens manufactured by LCD technology (LC Magna 3D printer model) were printed both on transversal and longitudinal directions as shown in Figure 1, in order to assess the effect of printing direction on mechanical performances.

Prior to performing thermomechanical analysis and static regime mechanical tests, all HTP and HTS samples were fully post-cured and cleaned according to the instructions in the datasheet. HTP material was post-cured for 1 h at 60 °C and HTS material for 2 h at 60 °C. Post-curing enabled parts to reach the highest possible strength and to become more stable. However, each resin behaved slightly differently when post-cured, and required different amounts of time and temperature to arrive at the material’s optimum properties. When a resin 3D-printed part finished printing, it remained on the build platform in a “green state”. This means that while parts have reached their final form, polymerization was not yet fully completed, and the part has yet to attain maximum mechanical properties. Post-curing with light and heat was key in unlocking this last mile of material properties for DLP 3D prints.

The Photocentric group provided in the corresponding datasheet the post-cure optimized settings for each individual resin marketed by them.

For the specimens manufactured by DLP technology, the internal structure was not shown, being an isotropic printing process. Instead, for the CFF manufactured specimens, the printing process was highly anisotropic. Thus, the internal structure is illustrated in Figure 2, extracted from Markforged X7 3D printing Eiger software used for slicing. For the ONYX specimens, the nozzle heating temperature was 275 °C, the thickness layer was 0.1 mm, and a 100% filling 32 layers printed at ±45 degrees according to Figure 2a were used. In the case of continuous carbon-fiber-reinforced ONYX_R samples, the nozzle heating temperature was 252 °C, the thickness layer was 0.125 mm, and one perimeter was applied in each tested specimen. For the tensile samples, 8 layers comprising 2 surface layers of ONYX ±45 degrees and 6 carbon fiber isotropic infill layouts were used as shown in Figure 2b (lower image) with the blue color, the fiber volume fraction being 87%. For the 3-point bending test samples, 32 layers comprising 2 surface layers of ONYX ±45 degrees and 30 carbon fiber isotropic infill layouts were used as shown in Figure 2b (upper image) with the blue color, the fiber volume fraction being 37%. The overall geometries for mechanical tests along with parameters optimized for the best printing results by using the Eiger software are summarized in Figure 2, as well as Table 1 and Table 2, respectively.

The nozzle temperatures and ONYX layer orientation were locked in for every printing process. When using continuous reinforcement, the layer thickness was also locked in, depending on the type of fiber used. This was done to ensure optimum results in the final printed part. For this reason, the number of layers was given by the sample thickness imposed by the ASTM standard. In the case of the DLP 3D printing technology, the printing parameters were also locked in, depending on the type of photopolymer resin used.

The tensile tests were performed on (L × W) 165 × 25 mm dog bone samples with 3.2 mm thickness, according to ASTM D638-14 for ONYX material, and rectangular samples of (L × W) 250 × 15 mm with 1 mm thickness, for the continuous reinforced composite (ONYX_R), according to ASTM D 3039 standard. A detailed drawing is available in Figure 3 for the flexural and tensile specimens.

Thermomechanical analysis (TMA) was performed on disc samples of 5 mm height and 9.8 mm diameter. The sample geometry, the internal architecture for the ONYX samples 3D printed using Markforged X7 by means of FDM (fused deposition modeling) technology, and carbon-fiber-reinforced ONYX_R samples 3D printed using Markforged X7 by means of CFF (continuous fiber fabrication) technology are shown in Figure 4a–c, respectively. ONYX samples were XZ printed with ±45° infill angle and 2 perimeters. Two samples were printed using layer thickness of 0.125 mm (40 layers) and 2 samples using 0.250 mm layer thickness (20 layers). The reinforced ONYX_R specimens were XZ printed 2 perimeters, 4 layers ±45° infill angle of ONYX and carbon fiber isotropic infill (layer thickness of 0.125 mm).

Dynamic mechanical analysis (DMA) experiments were performed on (L × W) 60 × 9 mm rectangular samples with 3 mm thickness. The internal architecture for the ONYX samples and for the carbon-fiber-reinforced ONYX_R samples are shown in Figure 5a,b, respectively.

A detailed drawing is available in Figure 6 for the TMA and DMA specimens.

### 2.2. DMA, TMA Analysis and Static Mechanical Test Methods

Dynamic mechanical analysis (DMA) experiments were performed according to ASTM D5023 standard [19] using a TA Instruments Discovery Series 850 dynamic mechanical analyzer, from TA Instruments, New Castle, USA. The elastic and viscoelastic behavior was determined in ”3-point bending mode” with the oscillation frequency set to 1 Hz, the amplitude set to 10 µm, using a preload force of 0.1 N, and on a temperature range from 30 °C to 200 °C using a heating rate of 2 °C/min.

Thermomechanical analyses (TMA) were performed following the procedure specified in the ASTM E831 standard [20] on the 3D-printed materials being studied, with the aim of measuring dimensional changes of solid materials as a function of temperature, time, or applied force. Coefficient of thermal expansion (CTE) and glass transition temperature (Tg) were determined and compared with DMA and CTE analyzer results. Analyses were performed using a TA Instruments Discovery TMA 450 analyzer, on a temperature range from 30 to 200 °C for HTS and HTP 3D-printed materials and from 30 to 160 °C for ONYX and ONYX_R using the same heating rate of 5 °C/min and a load of 0.1 N.

Static mechanical tests (tensile and 3-point bending) were performed using an Instron 3369 with 10 kN cell force, at room temperature. Extensometers were used to measure the strain in longitudinal and transverse directions to measure Poisson’s ratio for tensile samples. For the three-point bending test, the ASTM D790 standard was followed, using a 102.4 mm support span, depth of beam/support span ratio of 1/30, radius of 10.2 mm, crosshead rate of 5.46 mm/min, and midspan deflection 27.3 mm. All specimens were preloaded with 10N to assure contact between the specimen and the force application device. The same parameters were used for the reinforced specimens as for the unreinforced ones, the only difference being that for the reinforced specimens, a greater opening of the supports was used to reduce the effects of the interlayer shearing forces that could cause the failure of the specimen. For the tensile tests, two standards were used. ONYX, HTP, and HTS samples were tested using the ASTM D638-14 standard, using a length of 250 mm, an overall width bigger than the minimum specified (25 mm instead of 19 mm) in the considered standard because the clamping zone was enlarged to avoid failure in the clamping region, a length of 57 mm, and a width of 13 mm of the narrow section, with a gage length of 50 mm, using a 5 mm/min strain rate.

The carbon-fiber-reinforced ONYX_R samples were 165 mm by 25 mm by 1 mm, and tested following the ASTM D3039 standard, at room temperature, with displacement rate of 2 mm/min.

## 3. Results and Discussions

### 3.1. DMA Results

Figure 7 shows the average storage modulus E′, loss modulus E″, and tan δ of four samples for each of the four analyzed 3D-printed materials as a function of temperature.

Table 3 presents the statistical data obtained on the DMA samples.

Just before ONYX material reached 30 °C, it exhibited a stiffness higher than ABS, or conventional Nylon 6/6 polyamide [21]; the flexural modulus was 2.5 GPa, which also confirmed the datasheet value from the Markforged producer [22]. The loss modulus curves of ONYX show three different peaks corresponding to ά, α, and β transitions in order of decreasing temperature [21]. Current study results confirm the α transition just before 30 °C determining the Tg for ONYX material of 27 °C, coinciding with a significant decrease in modulus. This value matches with previously reported Tg for ONYX [23] and the datasheet, and was attributed to long-chain segmented motion within the main polymer chain [24]. The β transition occurring at negative temperatures between −60 °C and −70 °C [23] and the ά relaxation were previously observed in ONYX material, being associated with the mobility of interfacial amorphous phase in fiber-reinforced Nylon 66 composites [25]. Moreover, due to the presence of short carbon fibers, ONYX exhibits a higher storage modulus than Nylon 66 [23,26]. Dynamic mechanical analysis results for ONYX_R material are also displayed in Figure 7. For fiber-reinforced polymer composites, the dynamic mechanical properties depend on the fiber type, length, orientation loading, dispersion in matrix, and interaction between fiber and matrix. Continuous carbon-fiber-reinforced ONYX thermoplastic composite material (ONYX_R) presented a significantly higher storage modulus due to influence of continuous fibers, and hid the thermal properties of polymer matrices as it can be seen on the thermogram in Figure 7. The 25 GPa value is, nevertheless, lower than the one reported in the technical datasheet due to different design of the structural samples’ architectures (presence of walls, upper and lower ONYX layer, infill, and fiber orientation) within the preset study.

However, the ONYX_R showed completely different behavior. The storage modulus abruptly fell off at around 114 °C, corresponding to the maximum of the loss modulus and just after that of the tanδ curve, which confirmed the amorphous nature of the matrix of the ONYX_R material. This glass transition point at 114 °C was consistent with the one reported in [23] and was the only event recorded on the investigated temperature range with no melting or crystallization peaks. No literature data were reported previously for this HTP material, to our knowledge. The working temperature was above 23 °C since, below this temperature, the resin crystalizes. A peak of tan δ appeared around 173 °C for all four HTP material samples, marking the glass transition region where the material loses its stiffness. Furthermore, according to the manufacturer datasheet, the flexural modulus is 3.3 GPa, while in the present study, a higher 5 to 5.5 GPa was recorded for all tested samples. Across the Tg, the measured storage modulus (E′) decreased from the glassy plateau at approximately 5 GPa to the high-temperature rubbery plateau at about 500 MPa. The thermograms of HTP material were typical for an amorphous thermoplastic polymer, and showed the different states of the polymer behavior, as well as the beta transition temperature Tβ entering a glassy state, and the glass transition temperature Tg around 173 °C, followed by a decrease in both loss factor and storage modulus, possibly trailed by a rubbery elastic plateau and flow region. As for HTP material, no literature data were reported previously for this HTS material, to our knowledge. The working temperature was above 30 °C since, below this temperature, the resin crystalizes. Analyzing the loss factor curve, the transition temperature for entering a glassy state of Tβ was determined; furthermore, it was followed by the peak of tan δ corresponding to the glass transition temperature Tg around 157 °C, defined by a decrease in the storage modulus and material stiffness. The storage modulus evolution over the temperature range was similar to the one recorded on HTP material: it started at 5.5 GPa and achieved about 500 MPa at the glass transition region, and was lower than the 2.2 GPa stated by the material supplier in the datasheet.

### 3.2. TMA Results

Figure 8 shows the CTE mean curves of all the 3D-printed samples. Regarding ONYX material, based on nylon impregnated with short-chopped carbon fiber, the addition of these 2D fillers aims at maximizing the mechanical properties, the filament being more flexible and less brittle, and provides it with a means to adhere to the substrate. Nevertheless, these additions further complicate the thermal response of the material due to the filler’s ability to preferentially align along the print path. Therefore, understanding the thermomechanical response in terms of thermal relaxation/mobility, transitions (e.g., Tg), elastic modulus, and CTE for 3D-printed materials, both with and without filler and both with and without continuous reinforcement, is an important design parameter to achieve dimensionally accurate parts along with expected mechanical performances. There remain important areas in the design process that are less understood, such as the materials’ behavior and dimensional accuracy at elevated temperatures.

Table 4 presents CTE measurements for ONYX material obtained on the TMA-tested samples. The CTE values obtained on ONYX specimens printed in ZX orientation using ±45° infill angle (for both 125 µm and 250 µm layer thicknesses, respectively) was not highly dependent on temperature, averaging 3.13 × 10^−5^ 1/°C. Nevertheless, the layer thickness has influenced the dimensional behavior over the investigated temperature range.

The aim of these thermomechanical analyses on ONYX samples printed in ZX orientation with ±45° infill angle (the standard print setting in the Eiger Software is XY orientation) using two different 125 µm and 250 µm layer thicknesses, respectively, was to determine the influence of the printing layer thickness on the CTE. TMA results provided in Figure 9 display the values for the two series of ONYX material 3D printed using 125 µm and 250 µm layer thicknesses, clearly indicating a different evolution of the CTE over the investigated temperature range.

Nevertheless, both types of 3D-printed ONYX materials presented an increase in CTE after 30 °C and the same inflection point between 40 and 60 °C. Within this study, it was observed that CTE of ONYX (125 µm thickness layer) 3D printed was not highly dependent on temperature, averaging a 3.6 × 10^−5^ 1/°C value on the temperature range from 30 to 160 °C, confirming previously reported results [27,28]. Two distinct slopes in CTE measurements, typical for polymers, were observed for ONYX material; the linear coefficient of thermal expansion first slightly raised after 30 to 60 °C, from a value of 1.4 × 10^−5^ 1/°C to a peak of 3.5 × 10^−5^ 1/°C, where it remained almost constant up to 160 °C. On the contrary, ONYX samples developed using 250 µm thickness layer, showed a significant reduction in thermal expansion above 60 °C, from a value of 3 × 10^−5^ 1/°C to 3.5 × 10^−5^ 1/°C. These differences can be attributed to the lower number of interfaces and lower chopped carbon fiber content (associated to half of the number of layers compared to ONYX (125 µm layer) with a 3D-printed configuration), but also to stress relief after printing and improvements in crystallinity from the heat treatment [28].

ONYX_R material’s TMA curve from Figure 8 exhibited two inflection points. One was around 50 °C, where a slight increase in CTE was observed from 10.34 × 10^−6^ 1/°C (at 30 °C) to 1.49 × 10^−5^ 1/°C. A second transition point was recorded between 110 and 115 °C, where an increase in CTE was observed (1.72 × 10^−5^ 1/°C at 114 °C). This point also confirms the glass transition temperature Tg of 114 °C determined for ONYX_R material by means of DMA analysis. No clear softening points were detected as a negative deflection in dimension change on investigated materials. An overall CTE of 1.43 × 10^−5^ 1/°C was recorded over the temperature range from 30 °C to 120 °C. Nevertheless, a significant increase in CTE was observed after 120 °C from 31.9 × 10^−5^ 1/°C to 242.3 × 10^−6^ 1/°C at 156 °C. As mentioned earlier, ONYX_R samples were XZ printed isotropic with four upper and four lower ONYX layers of 45/−45° infill angle, and, additionally, four perimeters, using a 125 µm thickness layer. CTE of carbon fiber in the longitudinal direction was close to zero or negative; however, the overall CTE of the final part will be dependent on the infill orientation of the internal layers. Another factor that may contribute to enhanced expansion in the cross-flow direction was the potential for micro-porosity between adjacent printed beads, providing a buffer zone where the material can freely expand, mitigating the overall expansion observed at the macro scale [28].

### 3.3. Tensile and Three-Point Bending Mechanical Results

The average stress–strain curves of the three-point bending tests are illustrated in Figure 10 below, while Table 5 reports the values obtained for each three-point bending sample tested for each 3D-printed material investigated.

Three-point bending test results for the tested ONYX_R material specimens showed a mean flexural strength of 175 MPa with a standard deviation (SD) of 1.58 MPa. All flexural tested ONYX_R specimens elastically deform and fracture before deforming plastically: behavior specific to brittle polymers. All ONYX_R specimens had a brittle fracture characterized by a low elongation at the break and a sudden increase in stress during failure. The failure of the tested carbon-fiber-reinforced ONYX_R material specimens occurred suddenly, highlighted on the graph by a discontinuity (a sudden stress increase). This corresponds to the release of carbon fibers. The datasheet supplied by Markforged indicates a 540 MPa maximum flexural stress for the carbon fiber filament. It was mentioned that the specimens tested were particularly manufactured to obtain maximum values. For example, they were manufactured without the matrix layouts, by depositing carbon filament exclusively; this option was not accessible to general users.

Likewise, the maximum flexural stress was measured by a method similar to ASTM D790, requesting that the specimens were not tested up to failure before the end of the flexural test. In addition, previous reported results reveal that all printable materials have the tendency to absorb water from the environment, up to 8% of their weight [21,29]. During the printing process, the absorbed water creates bubbles and voids in the deposited filament, which weakens interlayer bonding of the final part. Hence, including a pre-drying step before printing might enhance mechanical performance of the final parts [30]. Furthermore, in all flexural tested ONYX_R specimens, it was observed that reinforced filament cross sections exhibit a high degree of inhomogeneity, with alternate polymer-rich and fiber-rich regions, most likely due to the filament fabrication method (Figure 11). The significant variation of fiber volume fraction could also lead to significant stress concentrations that can trigger premature failure of the materials. Nevertheless, this discrepancy, with respect to the producer’s datasheet, was previously reported by several studies. Parmiggiani et al. [31] reported an average 340.7 MPa maximum flexural stress for specimens with 56% continuous carbon fiber infill. Ghebretinase et al. [32] stated a value of the average maximum flexural stress of 270.7 MPa, where a test was performed according to the D7264 standard. Thus, for 14% infill decrease, the average maximum flexural stress decreased by 20.5%. In this paper, the average maximum flexural stress obtained for ONYX_R material was 175.75 MPa, which represents a 35% decrease for 5% less infill than reported in [32], and compared to the value provided by the Markforged datasheet, there was a 67% difference, but for 63% less infill, with differences attributed to all potential reasons mentioned above.

The failure of the ONYX test specimens did not occur until the test midspan deflection had been reached; therefore, the values presented are not actual breaking values. The values of the maximum stresses varied between 38 and 42 MPa, and showed good repeatability. The HTS resin shows a clear change in the average flexural strength depending on the printing direction as shown in Figure 10, from 78 MPa for the longitudinal printed specimens to 65 MPa for transversal printed specimens. In Figure 10 below, it is illustrated that the rupture occurred before the maximum value of the midspan deflection was reached. For HTP and HTS materials, no literature data were reported previously to our knowledge; nevertheless, results are slightly lower than values reported in the datasheet. The experimental data did not show large differences from one specimen to another, which indicated good repeatability of the additive manufacturing processes used in the production of the test specimens.

Regarding the continuous carbon-fiber-reinforced ONYX_R samples subjected to the flexural test, all the sources cited [31,32] present method differences such as higher infill percent, different testing method, different printing orientation and fiber direction, etc. Thus, these differences explain the lower value obtained for the average maximum flexural stress. Other factors that could affect/influence the obtained value are the temperature and humidity of the environment during the process (during manufacture, storage, and testing), since generally not all the parameters are measured and reported. A future study shall be conducted in order to determine the extent to which these factors may affect material properties. For example, in [33], it was exposed that the moisture and the print orientation considerably affect material properties. The material studied was ONYX FR, which was a flame-retardant variation of ONYX, with slightly better mechanical properties. The averaged stress–strain curves of the tensile test are illustrated in Figure 12 below, while Table 6 reports the values obtained for each tensile sample tested for each 3D-printed material investigated.

The average maximum tensile strength obtained from the tensile samples of the carbon-fiber-reinforced ONYX_R was 600 MPa with a standard deviation (SD) of 28.83 MPa and the average yield strain of approximately 0.025 mm/mm. The results are higher than other reported results for this type of material [32,34,35], but lower than the 800 MPa tensile strength indicated by the datasheet supplied by Markforged for the carbon fiber filament. It was mentioned that the specimens were specially prepared to obtain maximum values, but these fabrication options are not accessible to general users. M.J. Sauer observed in [16] that material properties are directly related to the number of carbon fiber strands loaded in tension within the part, and that the increase in material properties is linear. From the author’s best knowledge, there are no published papers that attained values close to the ultimate tensile strength provided by the Markforged datasheet. A reasonable explanation could be that the samples were manufactured entirely from carbon filament (100% carbon fiber reinforcement infill), as specified in datasheet. The stress increased because part of the load was taken from the reinforcing carbon fiber in ONYX_R specimens showing a brittle fracture behavior, characterized by a low specific elongation at break, observed at 2.5% strain, which is usual for reinforced composite materials.

The specimens fractured near the clamping section at both tab ends, but within the gauge length (Figure 13b). Thus, a double failure was observed on all samples, corresponding to MAT and MAB failure codes type according to the ASTM D 3039 standard. The carbon-fiber-reinforced ONYX_R specimens manufactured by CFF technology potentially present a weak bonding between the layers, especially between the matrix and reinforcement filaments; the printing orientations led to anisotropic material properties, as observed in this tensile test campaign, compared to other additive manufacturing technologies.

By examining the graph in Figure 12, it can be easily seen that, in the case of ONYX, the fracture occurred after it reached a level of plastic flow specific to the polymeric materials. The composition of ONYX (a mix of nylon and micro carbon fiber) was responsible for the results in the uniaxial tensile test graph. It was clear that local discontinuous reinforcement provided lower strength and stiffness compared to continuous reinforcement. ONYX samples showed a significantly lower behavior compared to reinforced ONYX_R specimens. The analysis of the obtained values led to an average value of 46 MPa for the maximum tensile test: higher than the value provided by the Markforged datasheet (37 MPa). This datasheet’s underestimation has been also noticed in other papers [31,36]. The composite laboratory of Aarhus University provided an extended datasheet of ONYX, based on their research [37]. It was interesting to observe how the tensile properties of ONYX were affected by the part printing direction. The specimens printed on the XY axis (longitudinal) obtained a lower tensile strength value (34.2 MPa) compared to the specimens printed on the XZ axis (53.6 MPa) [37]. The longitudinally printed specimen’s tensile strength [37] was slightly lower than the one provided by Markforged, and significantly lower than the results from this study. All tested ONYX specimens yielded after a separation plane, the tensile failure mode corresponding to AGM (angled gauge middle) according to the ASTM D3039 standard [38] (Figure 13). Such failure modes occurring at the gage region were previously reported for ONYX materials [30].

The values for the proportionality limit were close to those of the maximum tensile stress; materials exhibit a fragile behavior.

HTP and HTS 3D-printed materials showed reasonable performances in both tensile and three-point bending static regimes, higher than previously reported values for similar photopolymer resins [36]. Photocentric’s HTS 3D-printed material confirms the tensile elongation at break of 4.8% from the datasheet, although the ultimate tensile strength is 20% lower than reported values, possibly due to different printing parameters and testing methods. The flexural modulus was confirmed and the flexural strength was nearly the same as the value reported in the datasheet. HTS 3D-printed material displays high tensile strength and elongation comparable to acrylics and polyimides, providing minimal shrinkage and high-accuracy printed parts. HTP 3D-printed material is a temperature-resistant resin, with experimentally determined transition temperature around 175 °C and a deflection temperature around 230 °C, assuring both high strength and stiffness compared to other photopolymers [36,39]. Likewise, compared with ONYX material, both printing directions for HTP and HTS 3D-printed materials were stronger and stiffer under static mechanical solicitations. Regarding the effect of printing direction, no significant change was observed in HTP material between transversally and longitudinally printed samples in tensile regime. Nevertheless, for HTS material, the effect was visible, the tensile strength increasing from 45 MPa to around 65 MPa, the tensile strain from 2 to 4.8%, and the flexural strength rising from 65 MPa to 80 MPa while flexural strain stayed constant, when shifting the printing direction from transversal to longitudinal.

## 4. Conclusions

Both FFF and DLP additive manufacturing technologies investigated in this study showed great potential to be used as replacement for conventional methods (e.g., autoclave, VA-RTM) highly used in the aerospace industry for primary structures (e.g., fuselage), secondary structures (e.g., Outer Guide Vanes), or technological mold manufacturing. Likewise, the experimental data did not show a large dispersion, indicating good repeatability of the additive manufacturing processes used in the production of tested specimens. Within all investigated materials, the highest strength values were recorded by carbon-fiber-reinforced ONYX_R material regardless of the type of static testing regime. In addition, ONYX_R exhibited a glass transition point at 114 °C, which was consistent with previously reported values and comparable with usual CFRP composites developed by the abovementioned conventional methods. Thus, ONYX_R can be a potential candidate for aeronautic secondary structures by means of FFF additive manufacturing technology. Regarding the investigated photopolymers, results showed that HTS can be used for manufacturing low-temperature curing cycle molds (by means of DLP additive manufacturing technology), while HTP proved to be a good candidate for manufacturing molds necessary in the CFRP composite parts that are to be cured at higher temperatures, up to 230 °C, via conventional autoclave technology.

## Figures and Tables

**Figure 1 materials-15-05069-f001:**
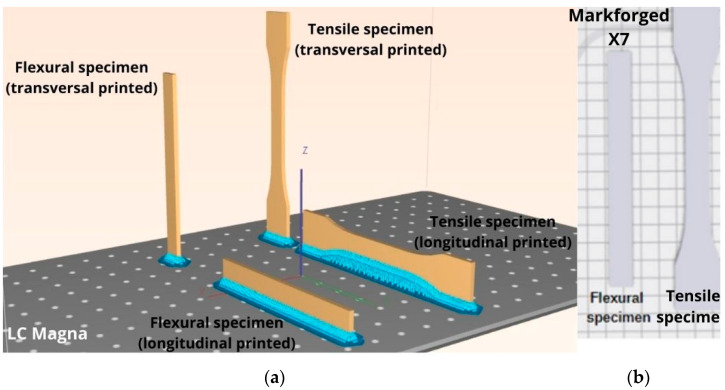
(**a**) Static mechanical test specimens’ printing directions on (**a**) LC Magna; (**b**) Markforged X7.

**Figure 2 materials-15-05069-f002:**
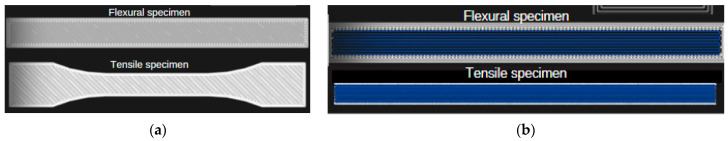
2D printing direction samples manufactured by CFF technology extracted from Eiger software: (**a**) ONYX; (**b**) continuous carbon-fiber-reinforced ONYX.

**Figure 3 materials-15-05069-f003:**
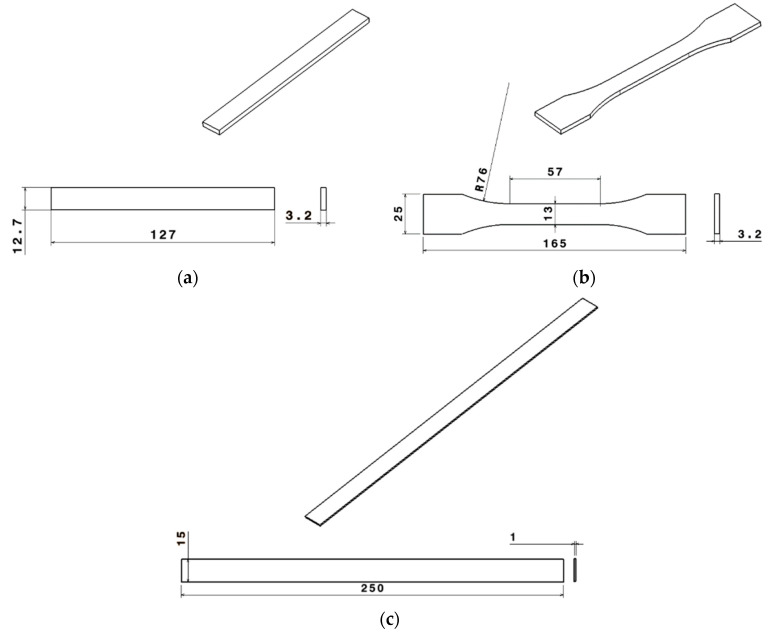
Specimens’ geometries according to standards: (**a**) three-point bending test; (**b**) tensile test—plastic; (**c**) tensile test—composite (reinforced ONYX).

**Figure 4 materials-15-05069-f004:**
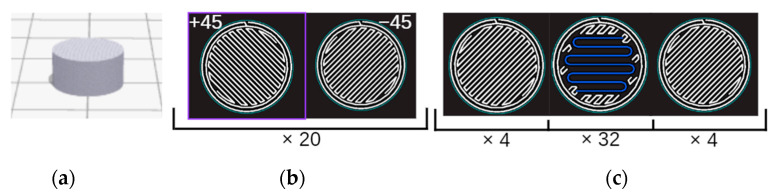
TMA sample details: (**a**) sample geometry; (**b**) ONYX sample architecture—40 layers of 0.125 mm thickness (alternating layers of ± 45 ONYX); (**c**) continuous carbon-fiber-reinforced ONYX sample architecture (blue—carbon filament reinforcement, white—ONYX matrix).

**Figure 5 materials-15-05069-f005:**
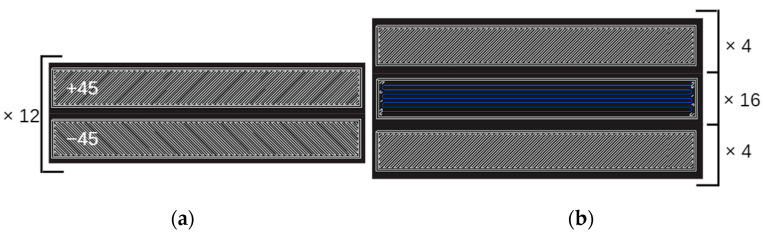
DMA sample details: (**a**) ONYX sample architecture—24 layers of 0.125 mm thickness (alternating layers of ± 45 ONYX); (**b**) continuous carbon-fiber-reinforced ONYX sample architecture (blue—carbon filament reinforcement, white—ONYX matrix).

**Figure 6 materials-15-05069-f006:**
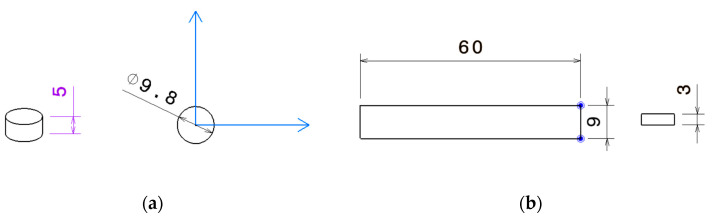
Sample geometry: (**a**) TMA; (**b**) DMA.

**Figure 7 materials-15-05069-f007:**
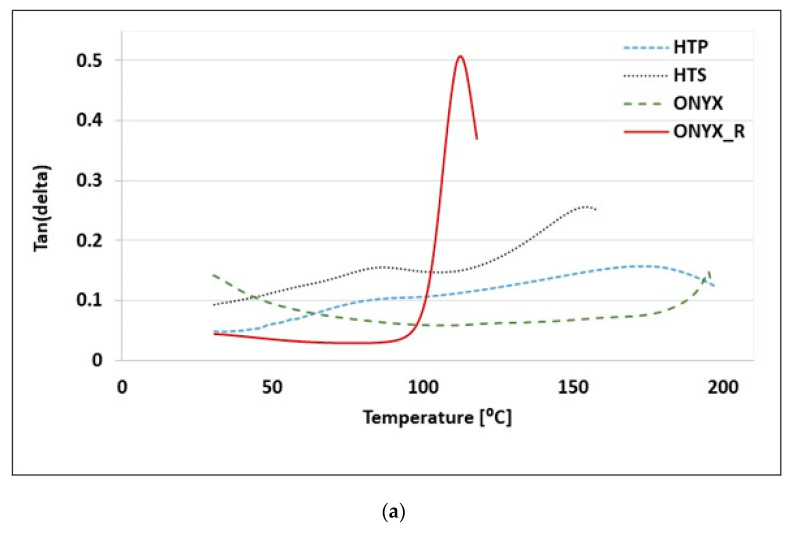
DMA measurements results: (**a**) Tan δ—Temperature curves; (**b**) Storage and Loss flexural modulus—Temperature curves.

**Figure 8 materials-15-05069-f008:**
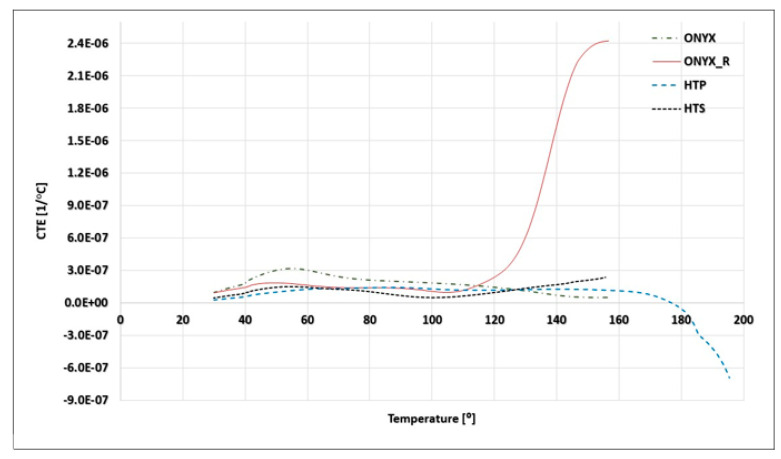
Thermomechanical analysis (TMA) curves showing the CTE of all investigated 3D-printed materials.

**Figure 9 materials-15-05069-f009:**
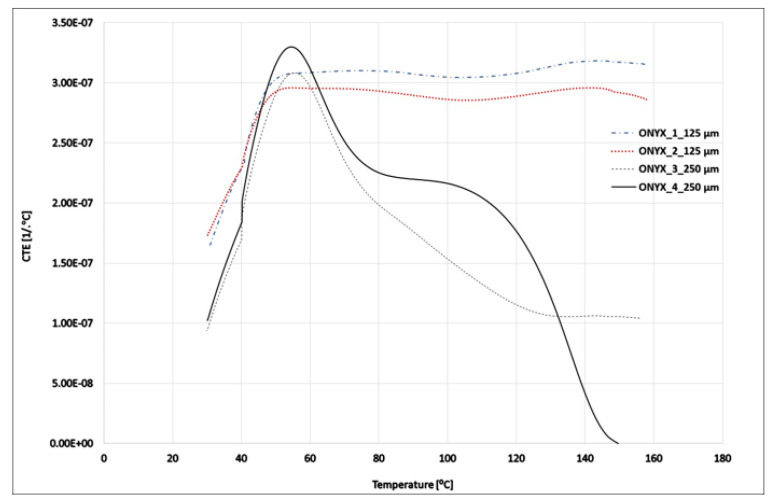
Thermomechanical analysis (TMA) curves showing the CTE the two series of ONYX material 3D printed using 125 µm and 250 µm layers.

**Figure 10 materials-15-05069-f010:**
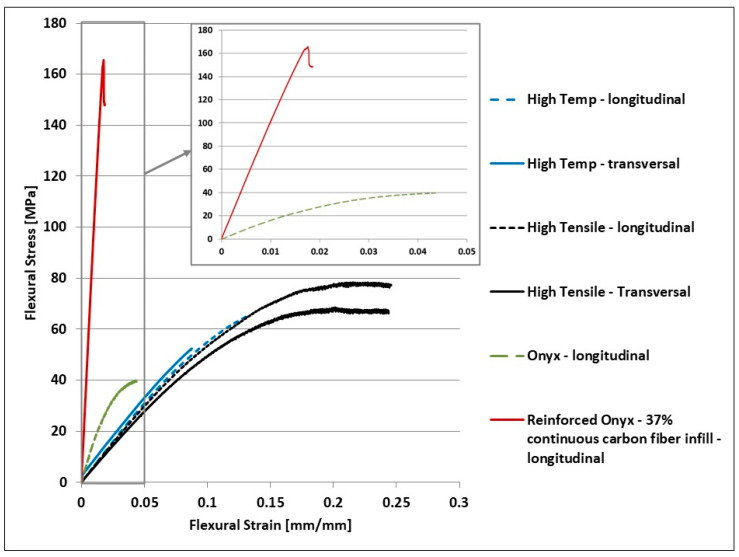
Average stress–strain curves in the 3-point bending flexural test.

**Figure 11 materials-15-05069-f011:**
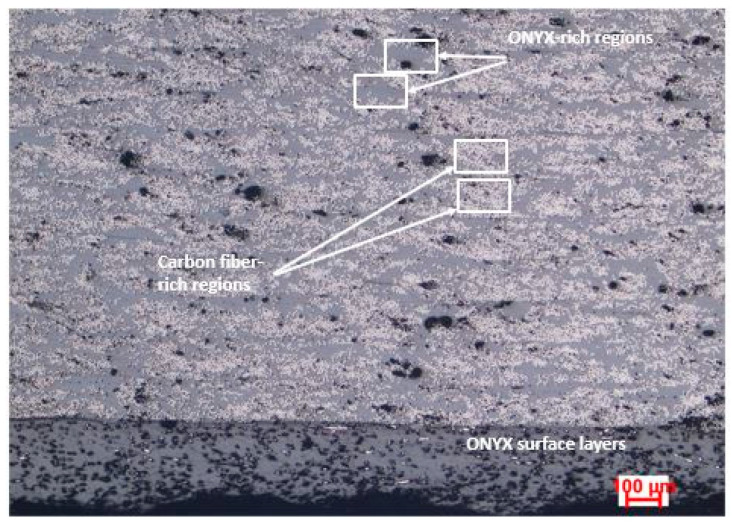
ONYX_R flexural tested sample cross section.

**Figure 12 materials-15-05069-f012:**
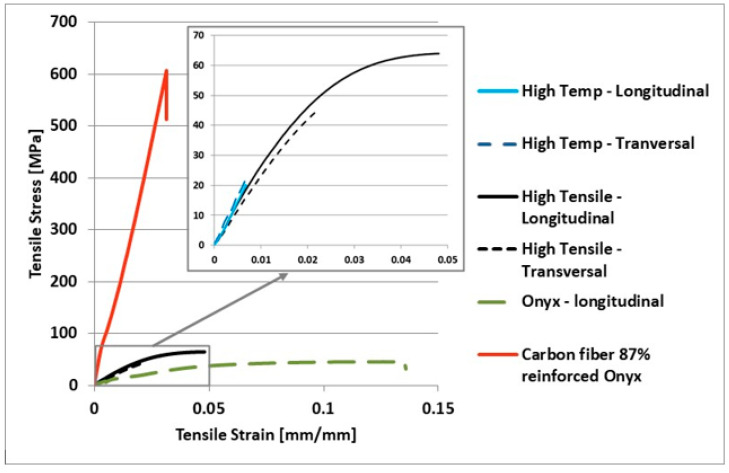
Stress–strain curves in the tensile test.

**Figure 13 materials-15-05069-f013:**
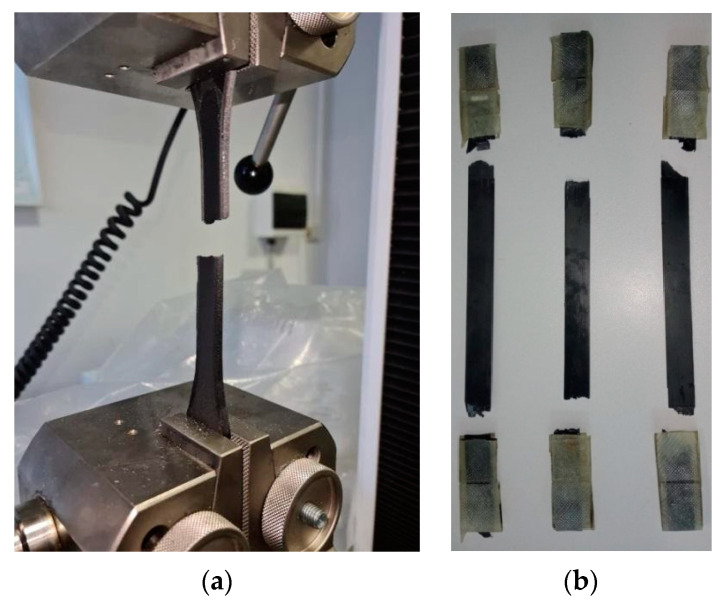
Samples at the end of the tensile test: (**a**) ONYX sample; (**b**) ONYX_R samples.

**Table 1 materials-15-05069-t001:** Reinforced ONYX Specimen printing parameters.

Reinforced ONYX Specimens					
ONYX Nozzle Temperature	Carbon Fiber Nozzle Temperature	Layer Thickness	Infill	ONYX Layer Orientation	Carbon Fiber Orientation
275 °C	252 °C	0.125 mm	100%	±45°	0°

**Table 2 materials-15-05069-t002:** ONYX Specimen printing parameters.

ONYX Specimens			
ONYX Nozzle Temperature	Layer Thickness	Infill	ONYX Layer Orientation
275 °C	0.1 mm	100%	±45°

**Table 3 materials-15-05069-t003:** DMA statistical data.

Material/Sample	ONYX	ONYX_R	HTP	HTS
Tan δ	0.1413 0.1785 - -	0.5208 0.4993 - -	0.1594 0.1589 0.1563 0.1482	0.2554 0.1467 0.2614 0.2527
Storage modulus [MPa]	2120.1 2487.1 - -	25,447 25,011 - -	6186.4 5466.4 55,439.4 5328	5597.1 5717.4 5370.8 5418
Loss modulus [MPa]	269.16 387.21 - -	4537.4 4562.7 - -	538.92 332.17 358.95 346.35	515.36 623.43 515.96 487.56

**Table 4 materials-15-05069-t004:** Overall TMA statistical data of CTE measurements for ONYX material (over 30 to 160° temperature range).

Material/Sample	ONYX	HTP	HTS
CTE [10^−6^ 1/°C]	318.4 (125 µm) 296 (125 µm) 308.4 (250 µm) 330 (250 µm)	147.4 144.5 141.3 -	271.7 265.3 177.3 -
Mean	313.2	144.4	238.1
SD	14.47	3.05	52.75
Std. Error	7.23	1.76	30.45

**Table 5 materials-15-05069-t005:** Maximum flexural strength results.

Material/Sample	ONYX	ONYX_R	HTS_L	HTS_T	HTP_L	HTP_T
Flexural Strength [MPa]	42.06	174.37	77.13	68.39	75.591	82.205
	41.47	176.08	77.48	64.84	68.976	70.866
	39.65	176.81	79.25	64.61	74.528	75.118
	39.32	-	82.56	64.84	72.165	-
	38.32	-	78.54	-	-	-
Mean	40.16	175.75	78.99	65.67	72.81	76.06
SD	2.42	1.58	4.69	3.3	8.6	32.81
CV [%]	6.03	0.9	5.94	5.03	11.81	43.14

**Table 6 materials-15-05069-t006:** Maximum tensile strength results.

Material/Sample	ONYX	ONYX_R	HTS_L	HTS_T	HTP_L	HTP_T
Tensile Strength [MPa]	46.23	621.9	61.16	58.06	27.36	36.61
	44.97	615.24	67.48	61.87	27.62	43.32
	49.84	622.63	68.77	52.54	23.79	35.06
	44.54	628.356	63.93	52.40	-	-
	44.51	-	-	53.65	-	-
Mean	46.02	622.03	65.33	55.7	26.26	38.33
SD	5.05	28.83	11.94	17.17	4.58	19.28
CV [%]	10.98	4.64	18.28	30.82	17.44	50.29

## Data Availability

Not applicable.

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
