# Peer review of "Investigation into Thermomechanical Response of Polymer Composite Materials Produced through Additive Manufacturing Technologies"

_materials, 2022, doi:10.3390/ma15145069_

Round 1

Reviewer 1 Report

The manuscript is ok, but it has several issues that need to be attended: 

line 17: ABS was not mentioned before

line 21: HTS was not previously mentioned or defined

line 98: The introduction is just focused on the AM fabrication methods... It does not cover important aspects of the relationship of the thermal properties and mechanical properties with the fabrication methodologies....

 In Materials and Methods is a big lack of  information such as statistical methodology, number of samples used, and techniques descriptions 

The manuscript has important issues in writing and grammar that need to be attended to e.g. verbal time mix, write in third singular person, wording, 

line 113: quality of the figures can be improved, also the concept of Magna was not mentioned in the text

line 118: sentence need no be revised

Check the spaces between the values and the units (in the whole manuscript) 

line 160: The dimensions for the DMA were the same as for the tensile test?
I think It is missing the sample dimensions 

line 170:It is better to use DSC for the determination of the Tg 

line 179: Please check the decimal divisor, sometime is comma, sometimes point

line 196: please, check the wording...

line 207: Please check expressions... What value are you referring to (flexural modulus?

line 209: These terms and their importance were not mentioned in the introduction as they are second order transitions. It is also important their relationship with the mechanical behaviour of the materials...

line 211: What is the importance of Tg while the focus of this work is mechanical testing?

line 234, 235, 237 please check the space between value and units

line 237: was Please, leave space between data and units in this case...

line 238: please check grammar

line 240: TGA was performed?

line 247: Please check the writing style...

line 252: statistical results are missing...

line 255: Figures 6, 7 and 8 are not mentioned in the text

line 257: redundant information

Figure 6: missing unit in the x axis

Figure 7 and 8: duplicate the data of fig 6?

line 271: if it was mentioned before, please no duplicate 

line 290: check writing

line 300: it seems to this belongs to the previous idea.. so, it should be in the previous paragraph

line 329 and 330: writing need to be improved

line 335: standard deviation is missing 

line 336: How many samples were tested? in the figure it seems to be just one.

line 337: Really?, which specimens? HTS looks like had plastic behaviour

line 337: had... the text should be writing in past tense

line 347-351: This topic is not connected with the mechanical behaviour that was discussed..
in addition, hygroscopic behaviour is well known, so this does not contribute too much to the state of art.

line 347: This is a new idea, so it should be a new paragraph.

line 355: microstructure? the term is not properly used here 

line 357: so, it is not new information... used other writing style...

please use period (point)... never mix with commas 

line 382: the word different is repeated unnecessarily 

line 396:What statistical methods were used?

how many samples were used

line 399: but it cannot compare the UTS of Carbon fiber to the composite...

line 404: this information should be in Material and Methods

line 423: should be table 2... this is no mentioned in the discussion of the results

line 443: positioning ???

Line 472: in prose... there is no limitation of the maximum number of words in this section???

The conclusions don't have references...

Check the references format

Author Response

Comments and Suggestions for Authors

The manuscript is ok, but it has several issues that need to be attended: 

line 17: ABS was not mentioned before ???

A: The full name of ABS is mentioned

line 21: HTS was not previously mentioned or defined

A: The full name of HTS and HTP is defined)

+ The abstract was modified to meet the condition of maximum 200 words

line 98: The introduction is just focused on the AM fabrication methods... It does not cover important aspects of the relationship of the thermal properties and mechanical properties with the fabrication methodologies....

 In Materials and Methods is a big lack of  information such as statistical methodology, number of samples used, and techniques descriptions 

The manuscript has important issues in writing and grammar that need to be attended to e.g. verbal time mix, write in third singular person, wording, 

A: We modified the introduction to meet the requirements of both reviewers.

line 113: quality of the figures can be improved, also the concept of Magna was not mentioned in the text

A: The concept of Magna is now introduced in the introduction section (see line 73-86)

line 118: sentence need no be revised

A: the sentence has been revised

Check the spaces between the values and the units (in the whole manuscript) 

A: Done

line 160: The dimensions for the DMA were the same as for the tensile test?
I think It is missing the sample dimensions 

A: Figure 6 was added

line 170:It is better to use DSC for the determination of the Tg 

A: The purpose is to determine the dynamic-mechanical response of the material, while the determination of Tg was a secondary investigation. We did find reports that show there is a difference in the Tg determined by DSC and DMA: 10.4028/www.scientific.net/AMM.719-720.91

line 179: Please check the decimal divisor, sometime is comma, sometimes point

A: Done

line 196: please, check the wording...

A: Done

line 207: Please check expressions... What value are you referring to (flexural modulus? (Am adaugat „flexural”).

A: It was specified in line 184 that the samples were tested using a thre point-bending clamp for flexural properties.

line 209: These terms and their importance were not mentioned in the introduction as they are second order transitions. It is also important their relationship with the mechanical behaviour of the materials

A: We feel like we have met the requirements by citing the references found in those lines.

line 211: What is the importance of Tg while the focus of this work is mechanical testing?

A: The paper presents both static mechanical behavior (tensile and flexural) and the dynamic thermo-mechanical properties (Tg and CTE) of four different materials. From the authors’ point of view, all the properties measured are important considering the application scope (to further address development of aeronautic secondary structures that are subjected to high temperatures, via 3D printed composite materials replacing conventional manufactured CFRP composites)

line 234, 235, 237 please check the space between value and units

A: Done

line 237: was Please, leave space between data and units in this case...

A: Done

line 238: please check grammar

A: Done

line 240: TGA was performed?

A: We do not have this type of equipment.

line 247: Please check the writing style...

A: We didn’t find anything.

line 252: statistical results are missing...?

A: Please see Tables 3&4

line 255: Figures 6, 7 and 8 are not mentioned in the text

A: Removed Figures 7 and 8 as they were splitted from figure 6. Figure 6 is mentioned in text, now.

line 257: redundant information

A: Deleted

Figure 6: missing unit in the x axis

A: Added

Figure 7 and 8: duplicate the data of fig 6?

A: Deleted

line 271: if it was mentioned before, please no duplicate 

A: Deleted

line 290: check writing

A: Checked

line 300: it seems to this belongs to the previous idea.. so, it should be in the previous paragraph

A: Done

line 329 and 330: writing need to be improved

A: Improved

line 335: standard deviation is missing 

A: See Tabel 5&6

line 336: How many samples were tested? in the figure it seems to be just one.

A: The average stress-strain curves are represented in Figure 8, and detailed data can be checked in Table 5

line 337: Really?, which specimens? HTS looks like had plastic behavior

A: We were reffering to ONYX_R. We clarified this in text.

line 337: had... the text should be writing in past tense

A: Done

line 347-351: This topic is not connected with the mechanical behaviour that was discussed..
in addition, hygroscopic behaviour is well known, so this does not contribute too much to the state of art.

A: The intention was to highlight previous reported data from other researchers that motivate the difference in the mechanical behaviour of the present study tested samples, when compared to the manufacturer datasheet of the material.

line 347: This is a new idea, so it should be a new paragraph.

A: Done

line 355: microstructure? the term is not properly used here 

A: Changed

line 357: so, it is not new information... used other writing style...  

A: It is our finding and we think it is relevant as we did not find any connections highlighted in other papers.

please use period (point)... never mix with commas 

A: Done

line 382: the word different is repeated unnecessarily 

A: Deleted

line 396:What statistical methods were used? how many samples were used.

A: The number of samples and methodology was done according to every ASTM standard that was mentioned for every mechanical test.

line 399: but it cannot compare the UTS of Carbon fiber to the composite...

A: Yes, we totally agree. But at the same time the technology is much faster and the geometries you can produce are of a higher degree of complexity. It is always pros and cons.

line 404: this information should be in Material and Methods

A: Moved

line 423: should be table 2... this is no mentioned in the discussion of the results

A: Changed

line 443: positioning ???

A: We will call it orientation from now on.

Line 472: in prose... there is no limitation of the maximum number of words in this section???

A: We revised the conclusions

The conclusions don't have references...

A: Done

Check the references format

A: This section was recreated.

Reviewer 2 Report

The authors provided a manuscript on additive manufactured polymer composites subjected to static mechanical tests and dynamic mechanical analysis. The comments for the paper are listed below:

1) The title of the manuscript is a bit confusing. "Investigation into static and dynamic mechanical response" suggests that materials will be tested at low (static) and high strain rates (dynamic mechanical response). Since the dynamic mechanical response is dedicated to high strain rate tests such as SHPB, the title might mislead the readers. Please correct the title.

2) Abstract: while using abbreviations for the first time (CFRP, ABS, HTP, HTS), please consider using full names while mentioning them for the first time.

3) Introduction: this section is too general. The well-known information about additive manufacturing technologies should be replaced by current achievements in 3d printing of investigated materials. The introduction should cover the up-to-date knowledge on the materials and technologies in question. The current version of the abstract is based on outdated papers, technical data sheets, and standards mainly, thus the reviewer feels that it is not reflecting the background sufficiently. Please revise, provide a knowledge gap and the main motivation of the paper.

4) Materials and fabrication methods: Please summarize the optimized printing parameters for each material in form of the table.

5) Materials and fabrication methods: Please provide the engineering drawing of the manufactured and subsequently mechanically tested specimens.

6) Materials and fabrication methods: Line 116 "HTP material was post cured for 1hr at 60°C and HTS material for 2 hours at 60°C" - please explain why the material was heat treated.

7) Materials and fabrication methods: Lines 122-125: how the printing parameters were determined?

8) Materials and fabrication methods: Lines 126-130, 142-143: how the number of layers was determined?

9) DMA results: Figure 5: would it be possible to use GPa instead of MPa?

10) DMA results: lines 214 - 216: "the β transition occurring at negative temperatures between -60°C and -70°C" -  Since the tests are performed in the temperature range from 30 to almost 200C, what is the point of providing such information? 

11) DMA results: Line 224-227: if the 25GPa result was lower than this reported in the technical data sheet, why the similar or the same specimen design was not used to validate the data? It seems like the research methodology was not planned sufficiently well.

12) DMA results: Line 236: How explain the differences between obtained results and those provided by the material supplier?

13) TMA results: Lines 257 - 261: "Moreover, the build orientation is a key driver in the final thermal and mechanical performances of the materials, explaining the difference in results with respect to the producer data sheets and aligning well with previously reported research on the mechanical properties of 3D printed parts [20-24]." - each reference should be discussed separately.

14) TMA results: Lines 271-272: why such an infill angle and 125um layer was used to manufacture ONYX samples?

15) TMA results: Lines 310 - 313, 324 - 327: the references to support such statements are required.

16) Tensile and three-point bending tests: Figure 10: why the bending curves do not start from 0,0 point?

17) Tensile and three-point bending tests: Lines 352-354: "Furthermore, in all flexural tested ONYX_R specimens it was observed that reinforced filament cross sections exhibit a high degree of inhomogeneity, with alternate polymer-rich and fibre-rich regions, most likely due to the filament fabrication method." - the fracture surfaces should be provided to support such statement.

18) Tensile and three-point bending tests: Please double-check the values of SD and CV in Tables 1-2 (for example Table 1, the SD value for HTP_T). The values seem not correct.

19) Tensile and three-point bending tests: Lines 380-390: Since the temperature and the humidity are affecting the mechanical response of manufactured specimens, how ensure that 3d printed components will possess the required properties?

20) Tensile and three-point bending tests: Lines 415 - 416: Since the fracture occurred near the clamping sections it seems that inadequate clamping force was used during the testing. Please discuss.

21) Conclusions: The conclusions should be short and objective and not the description of results, nor should they be a summary.

The manuscript submitted has some publishing potential however, the issues raised by the reviewer should be explained. The reviewer will reconsider this paper after major revision.

Author Response

1) The title of the manuscript is a bit confusing. "Investigation into static and dynamic mechanical response" suggests that materials will be tested at low (static) and high strain rates (dynamic mechanical response). Since the dynamic mechanical response is dedicated to high strain rate tests such as SHPB, the title might mislead the readers. Please correct the title.

A: We revised the title

2) Abstract: while using abbreviations for the first time (CFRP, ABS, HTP, HTS), please consider using full names while mentioning them for the first time.

A: The title is changed

3) Introduction: this section is too general. The well-known information about additive manufacturing technologies should be replaced by current achievements in 3d printing of investigated materials. The introduction should cover the up-to-date knowledge on the materials and technologies in question. The current version of the abstract is based on outdated papers, technical data sheets, and standards mainly, thus the reviewer feels that it is not reflecting the background sufficiently. Please revise, provide a knowledge gap and the main motivation of the paper.

A: We rethought the introduction

4) Materials and fabrication methods: Please summarize the optimized printing parameters for each material in form of the table.

A: Please check Table 1&2

5) Materials and fabrication methods: Please provide the engineering drawing of the manufactured and subsequently mechanically tested specimens.

A: Please check Figure 3 & 6

6) Materials and fabrication methods: Line 116 "HTP material was post cured for 1hr at 60°C and HTS material for 2 hours at 60°C" - please explain why the material was heat treated.

A: Please, see lines 106-115

7) Materials and fabrication methods: Lines 122-125: how the printing parameters were determined?

A: The parameters are optimized for the best printing results by the software provider.

8) Materials and fabrication methods: Lines 126-130, 142-143: how the number of layers was determined?

A: We followed the ASTM standard for each mechanical test. We had the thickness imposed by the standard. Further explanations are added in text (lines 141-147)

9) DMA results: Figure 5: would it be possible to use GPa instead of MPa?

A: The rest of the plots are in MPa. If it would be possible, we would like to keep it in MPa. If not, we will change.

10) DMA results: lines 214 - 216: "the β transition occurring at negative temperatures between -60°C and -70°C" -Since the tests are performed in the temperature range from 30 to almost 200C, what is the point of providing such information? 

A: We considered that the information is important, as some aerospace applications meet negative temperatures in operational conditions. This could provide the reader important information.

11) DMA results: Line 224-227: if the 25GPa result was lower than this reported in the technical data sheet, why the similar or the same specimen design was not used to validate the data? It seems like the research methodology was not planned sufficiently well.

A: The manufacturer provided the flexural modulus based on different standards for 3 point bending. What we obtained was for DMA testing where the samples are much smaller and given a smaller area of reinforcement, the difference is expected due to printer limitations most probably. We also conducted mechanical tests according to the standard used by the manufacturer and we obtained better results. The manufacturer provided the best results its material can obtain in optimal conditions which can not be always obtained due to different applications. We tested samples in a configuration that would provide the best results in surface quality, dimensional accuracy and mechanical strength for the aeronautical secondary structures we want to develop in the future.

12) DMA results: Line 236: How explain the differences between obtained results and those provided by the material supplier?

A: The results here are obtained on DMA, different from the mechanical tests conducted by the manufacturer. The DMA required smaller sized samples which could be a decisive factor in obtaining a higher value of 5.5 GPa.

13) TMA results: Lines 257 - 261: "Moreover, the build orientation is a key driver in the final thermal and mechanical performances of the materials, explaining the difference in results with respect to the producer data sheets and aligning well with previously reported research on the mechanical properties of 3D printed parts [20-24]." - each reference should be discussed separately.

A: We were asked by the other reviewer to eliminate de information which was considered “redundant”

14) TMA results: Lines 271-272: why such an infill angle and 125um layer was used to manufacture ONYX samples?

A: The manufacturer limits the use of some parameters, to obtain the optimum results. More information can be checked in line 141-147.

15) TMA results: Lines 310 - 313, 324 - 327: the references to support such statements are required.

A: We forgot to do that, now its there. Thank you.

16) Tensile and three-point bending tests: Figure 10: why the bending curves do not start from 0,0 point?

A: This happened due to the initial pre-loading.

17) Tensile and three-point bending tests: Lines 352-354: "Furthermore, in all flexural tested ONYX_R specimens it was observed that reinforced filament cross sections exhibit a high degree of inhomogeneity, with alternate polymer-rich and fibre-rich regions, most likely due to the filament fabrication method." - the fracture surfaces should be provided to support such statement.

A: Please, see Figure 11

18) Tensile and three-point bending tests: Please double-check the values of SD and CV in Tables 1-2 (for example Table 1, the SD value for HTP_T). The values seem not correct.

A: We double checked. The formulae is precisely applied as stated in the ASTM standard. This is due to the different behaviour of the resin material (brittle). It basically explodes when reaching total failure.

19) Tensile and three-point bending tests: Lines 380-390: Since the temperature and the humidity are affecting the mechanical response of manufactured specimens, how ensure that 3d printed components will possess the required properties?

A: It was not the aim for the present study. We plan to look into different surface coatings and make a different research paper on this subject.

20) Tensile and three-point bending tests: Lines 415 - 416: Since the fracture occurred near the clamping sections it seems that inadequate clamping force was used during the testing. Please discuss.

A: Please, see lines 421-424

21) Conclusions: The conclusions should be short and objective and not the description of results, nor should they be a summary.

A: We have revised the conclusions.

The manuscript submitted has some publishing potential however, the issues raised by the reviewer should be explained. The reviewer will reconsider this paper after major revision.

Round 2

Reviewer 1 Report

The authors put efforts to modify the manuscript. 

Nevertheless, it is necessary to revise all the work to check details as: 

1. spaces between the value and the units usually is for almost all the cases, except for % °, °C (for these cases no spaces are required)

2. present tense can be used according to text that is written, but is highly recommended to use past in all the text.  The whole manuscript needs to be revised. It is highly recommended a philological review.

3. Quality of figures need to be improved. For example, dimensions inside figure 3 are barely legible. In figure 9 it is difficult to observe the line for ONYX_3. Also, in fig. 9 the value format Y axis needs to be checked. It is better to use the same format for all the axes.  

4. table 4 title is TMA statistical data, but in there is omitted even the standard deviation. The authors mentioned that they follow the ASTM, however statistical analysis is needed e.g. ANOVA, experimental design, etc.

5. Figure 5 shows the fracture of the sample, but poor information is bringing without SEM micrograph

6. the conclusion section is too long.

Author Response

Comments and Suggestions for Authors

The authors put efforts to modify the manuscript. 

Nevertheless, it is necessary to revise all the work to check details as: 

  1. spaces between the value and the units usually is for almost all the cases, except for % °, °C (for these cases no spaces are required)

  1. present tense can be used according to text that is written, but is highly recommended to use past in all the text.  The whole manuscript needs to be revised. It is highly recommended a philological review.

  1. Quality of figures need to be improved. For example, dimensions inside figure 3 are barely legible. In figure 9 it is difficult to observe the line for ONYX_3. Also, in fig. 9 the value format Y axis needs to be checked. It is better to use the same format for all the axes.  

  1. table 4 title is TMA statistical data, but in there is omitted even the standard deviation. The authors mentioned that they follow the ASTM, however statistical analysis is needed e.g. ANOVA, experimental design, etc.

  1. Figure 5 shows the fracture of the sample, but poor information is bringing without SEM micrograph

  1. the conclusion section is too long.

Submission Date

10 June 2022

Date of this review

04 Jul 2022 21:08:08

Answers to second Review from 04 Jul 2022 21:08:08

  1. We revised the manuscript according to indications
  2. We revised the manuscript according to indications
  3. Figure 3 and Figure 9 were changed.
  4. We revised Table 4 according to indications
  5. The fractured samples from Figure 13 were classified according to ASTM D3039 standard. Figure 11 was changed and provide the SEM micrograph of ONYX_R cross section flexural tested samples
  6. The Conclusion section was additionally shortened since the first revision.

Submission Date of answers to the second review: 07.07.2022

Reviewer 2 Report

The paper has been improved at some point but there are still some issues. Further comments on the manuscript:

1) Although the introduction was rewritten, it remained too general. The literature review, which is based on 8 papers from which 5 of them were discussed by referring [1-5], is not the proper one.

2) Line 144 - "The Photocentric group developed an in-house post-cure study to identify optimized settings for each individual resin marketed by them." - any reference? If this study was not published, then it should be briefly introduced here.

3) The authors compare their results many times to those provided by the supplier/manufacturer even though the specimen geometry or the conditions used throughout the tests are different. Please clearly state in the paper the purpose of doing such comparisons. It feels like the authors want to double-check the datasheets provided.

4) Please work on graphs. Keep the consistency of the format including the same size of the font, the number format etc.. Make them look professional.

5) Bending curves - even if the pre-loading occurs, the lines should start from 0,0. Please correct.

6) Answer 17 / A: Please, see Figure 11 - the figure is not referred in the text... Furthermore, this figure does not support what the authors claimed.

These comments must be included to reconsider the manuscript for publication.

Author Response

Comments and Suggestions for Authors

The paper has been improved at some point but there are still some issues. Further comments on the manuscript:

1) Although the introduction was rewritten, it remained too general. The literature review, which is based on 8 papers from which 5 of them were discussed by referring [1-5], is not the proper one.

2) Line 144 - "The Photocentric group developed an in-house post-cure study to identify optimized settings for each individual resin marketed by them." - any reference? If this study was not published, then it should be briefly introduced here.

3) The authors compare their results many times to those provided by the supplier/manufacturer even though the specimen geometry or the conditions used throughout the tests are different. Please clearly state in the paper the purpose of doing such comparisons. It feels like the authors want to double-check the datasheets provided.

4) Please work on graphs. Keep the consistency of the format including the same size of the font, the number format etc.. Make them look professional.

5) Bending curves - even if the pre-loading occurs, the lines should start from 0,0. Please correct.

6) Answer 17 / A: Please, see Figure 11 - the figure is not referred in the text... Furthermore, this figure does not support what the authors claimed.

These comments must be included to reconsider the manuscript for publication.

Submission Date

10 June 2022

Date of this review

04 Jul 2022 13:51:26

Answers to second Review from 04 Jul 2022 13:51:26

  1. The introduction was revised according to indications. Supplementary literature research studies results and references were integrated in the Introduction section. Nevertheless, literature data related to specific additive manufacturing materials such as HTP and HTS photopolymers availability is limited.

  1. Regarding Lines 114,115 referring to the text "The Photocentric group developed an in-house post-cure study to identify optimized settings for each individual resin marketed by them", the authors would like to acknowledge that the information came from the technical data sheet (https://photocentricgroup.us/product/hightemp-dl400/; https://photocentricgroup.com/product/uv-high-tensile-resin/ of the Photocentric group manufacturer regarding the specific post-cure treatment for each of under study materials: HTP and respectively HTS. Within the present study no changes of these settings or post-cure treatments were performed.

  1. Yes, one of the interest of the paper targeted a comparison of the present study experimental data with the optimum corresponding parameters reported by the supplier/manufacturer, the raison being to underline the necessity of experimental data for different configurations or applications that the optimum ones reported in the technical data sheet.

The technical data sheet refers to reinforced parts (e.g. in the case of Markforged technology the data from the supplier is given for 100% carbon fiber reinforcement), nevertheless, this reinforcement option is not available for general users or even if available, cannot be used for complex geometries parts where the adhesion is ensured by using Onyx matrix in combination with continuous fiber reinforcement witch never achieves 100% of carbon fiber reinforcement).

  1. The format including the same size of the font, the number format of the graphs were modified according to the requirement.

  1. The Bending curves were modified in order to start from 0,0 as required.

  1. Figure 9 was deleted from the previous 1st reviewed text (line 386) since it was not linked to the quoted position and related text.

Regarding three-point bending tests text paragraph (from 411-414 lines) the figure 11 (line 427) image was changed in supporting the text claims.

Figure 11. ONYX_R flexural tested sample cross section

Submission Date of answers to the second review: 07.07.2022

Round 3

Reviewer 1 Report

The text still is written mixing present and past (specially in materials and methods), but the whole work should be revised.

It is confusing to use old and new figures. In some cases, it is not possible to see the improvements.

Figure 11 is now divided into two? If so, you forgot the divisor a) and b). it seen that the SEM micrograph has low quality. Maybe it is because the quality of the sent version to be revised.

the figures in a manuscript usually have the same size, but in this case, the figures have big variations.

Author Response

Answers to Comments and Suggestions for Authors (review 3)

  1. The text still is written mixing present and past (specially in materials and methods), but the whole work should be revised.

Answer: The overall text was revised according to the requirement.

  1. It is confusing to use old and new figures. In some cases, it is not possible to see the improvements.

Answer: All figures were revised according to the requirement.

  1. Figure 11 is now divided into two? If so, you forgot the divisor a) and b). it seen that the SEM micrograph has low quality. Maybe it is because the quality of the sent version to be revised.

Answer: No, it is a single picture in figure 11 (showing explicitly the two regions of rich matrix and rich carbon fibers areas, so no need to add divisor a) and b).

Figure 11 original file is attched in the platform with the new manuscript, see also below the original.

the figures in a manuscript usually have the same size, but in this case, the figures have big variations.

Answer: In this paper we chose to add different sizes to the image/graphs corresponding figures depending on the level of details that each  image contained.

Date of answers to review 3: 11.07.2022

Reviewer 2 Report

The revisions have been made. The paper can be considered for publication by the editor.

Author Response

Thank you!